# Multi-modal Auto-regressive Modeling via Visual Tokens

## ABSTRACT

Large Language Models (LLMs), benefiting from the auto-regressive modelling approach performed on massive unannotated texts corpora, demonstrates powerful perceptual and reasoning capabilities. However, as for extending auto-regressive modelling to multimodal scenarios to build Large Multi-modal Models (LMMs), there lies a great difficulty that the image information is processed in the LMM as continuous visual embeddings, which cannot obtain discrete supervised labels for classification. In this paper, we successfully perform multi-modal auto-regressive modeling with a unified objective for the first time. Specifically, we propose the concept of visual tokens, which maps the visual features to probability distributions over LLM's vocabulary, providing supervision information for visual modelling. We further explore the distribution of visual features in the semantic space within LMM and the possibility of using text embeddings to represent visual information. Experimental results and ablation studies on 5 VQA tasks and 4 benchmark toolkits validate the powerful performance of our proposed approach.

## CCS CONCEPTS

• **Computing methodologies → Natural language generation**; **Image representations**.

## KEYWORDS

Multimedia Foundation Models, Multi-modal Auto-regressive Modeling, Vision and Language, Modal Interpretation

## 1 INTRODUCTION

Over the past year, Large Language Models (LLMs) have made impressive breakthroughs and successfully use language as a common interface for a wide variety of real-world tasks. Benefiting from the auto-regressive modelling approach performed on massive unannotated texts, LLM is able to learn general-purpose semantic information and powerful reasoning capabilities from natural language corpora. The success of LLMs attracts researchers to explore Large Multi-modal Models (LMMs), which aim to extend the powerful text-only perceptual and reasoning capabilities of LLMs to scenarios dealing with multi-modal inputs. However, as for extending auto-regressive modelling to multi-modal scenarios, there lies a great difficulty that the image information is processed in the LMM as continuous visual embeddings, which cannot obtain

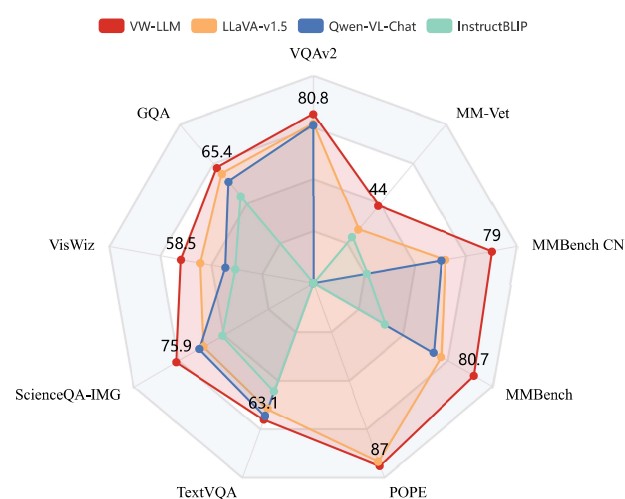

**Figure 1: Performance of our proposed VT-LMM. We set the unreported results in the original paper to 0 to avoid confusion.**

discrete supervised labels for classification. As a compromise solution, mainstream LMMs choose to compute losses only for the language portion of multi-modal interleaved sequence. As illustrated in figure 2(a), the training objective of mainstream methods [2, 4, 14, 19, 22] focuses on predicting language responses in multi-modal contextual sequences that depend on visual information, where the visual information merely acts as contextual cues and does not serve as supervision. This unfair treatment of different modal information in multi-modal sequences lacks the process of learning different modal information utilising the inference capabilities of the LLM, severely limiting the potential of LMM and resulting in under-utilisation of the training data. Although recent works [26, 27] propose to unlock the LLM for text pre-training by using a regression task to predict the value of next visual feature in the pre-training phase (Fig. 2(b)), the inconsistent optimisation goals of its visual and linguistic components are not conducive to unified multi-modal auto-regressive modelling. [3] has also proposed learning visual features using an auto-regressive classification tasks, but it uses a pre-trained image tokenizer, such as VQVAE or VQGAN, to cluster image features into a grid of discrete tokens which does not combine discrete visually supervised information with LLM.

In this paper, we successfully perform multi-modal auto-regressive modeling with a unified objective for the first time. Specifically, we introduce the concept of **visual tokens** to construct representations of visual features in the language semantic space inside the LMM, thus implementing visual modelling of LMM in the form of classification rather than regression. We also propose the **V**isual **T**oken guided **L**arge **M**ulti-modal **M**odel (VT-LMM), a novel multi-modal model that inherits the successful learning paradigm of LLM in the pre-training task, i.e., predicting the next image/text token

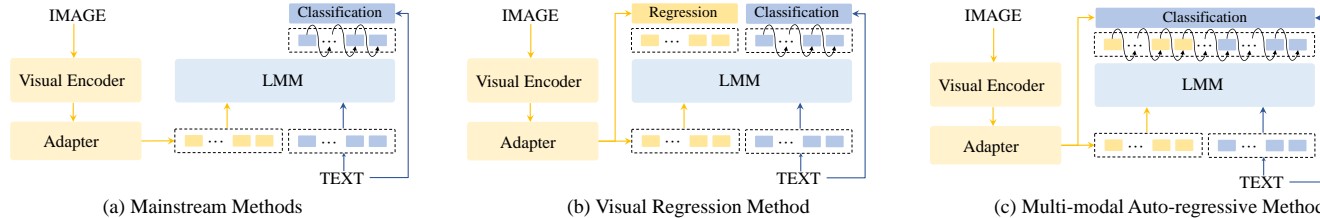

Figure 2: The comparisons between different LMMs. (a) Mainstream methods focuses on predicting language responses in multi-modal contextual sequences that depend on visual information, where the visual information merely acts as contextual cues and does not serve as supervision. (b) Visual regression method takes regression task to predict the value of next visual feature and performs joint training with text. (c) Our multi-modal auto-regressive method use visual tokens to construct visual supervision labels and enabling multi-modal auto-regressive modelling with unified classification objective.

in an auto-regressive manner. With the help of visual tokens, it's possible to train LMM with auto-regressive modeling without any specific architectural modifications, as shown in figure 2(c). Just as the LM head component is used in LLM to accomplish the mapping of text features to interpretable semantics, we correspondingly construct the VM head component, using the same structure, intent on obtaining the representation of each visual embedding in the language semantic space of LLM. For each visual embedding, we use the VM head to map it into probability distributions over the pre-trained vocabulary, which we call visual tokens. After that, the visual tokens corresponding to the image modal and the one hot label corresponding to the text modal are intertwined to form the supervised information for multi-modal auto-regressive modelling. Further, the pre-trained embeddings weights of LLM can be approximated as a set of complete bases of the language semantic space, i.e., the semantic information covered by the embeddings can basically cover the whole semantic space. Therefore, we further explored whether pseudo image features constructed with visual tokens and LLM's pre-trained embeddings can convey visual information to the model. We conducted extensive experiments on five commonly used visual question answering benchmarks and four LMM-evaluating benchmark toolkits, and the experimental results demonstrate that our VT-LMM, by constructing visual tokens to introduce visual supervisory information, achieves best performance among models of the same scale, and obtains vision-language understanding capability competitive to or even surpassing that of 13B or even larger scale models with a scale of 7B. Our main contributions are as follows:

- We propose the concept of visual tokens, which maps visual features into language semantic space, enabling LMM to perform auto-regressive modelling over multi-modal sequence..
- We further explored the distribution of visual features in the semantic space within the LMM and the possibility of using text embeddings to represent visual information.
- Experimental results and ablation studies on 5 VQA tasks and 4 benchmark toolkits validate the powerful performance of our proposed approach.

## 2 METHOD

### 2.1 Multi-modal Learning

To extend the powerful text-only perceptual and reasoning capabilities of LLM to scenarios dealing with multi-modal inputs, existing multi-modal learning method of LMM typically use the adapter structure to transform the visual features encoded by pre-trained visual backbone into the semantic space of LLM and construct multi-modal input sequences together with text embeddings. The adapter could be well-designed visual resampler [2, 4], Multi-layer Perceptron (MLP) [19] or even simple Linear layer [20].

Specifically, given an image and the corresponding text instruction, the multi-modal input sequence $X_{\text{input}}$ of LMM is constructed as follows:

$$
\begin{aligned}
X_{\text{image}} &= \text{AD}(\text{VE}(\text{image})), \\
X_{\text{text}} &= \Psi(\text{text}), \\
X_{\text{input}} &= [x_0^m, x_1^m, \ldots, x_L^m], \\
m &\in \{v, t\}, \ x^v \in X_{\text{image}}, \ x^t \in X_{\text{text}},
\end{aligned}
\tag{1}
$$

where $L$ represents the length of multi-modal input sequence, $\Psi$ represents the embedding layer of LLM, VE represents the visual encoder and AD represents the adapter.

Subsequently, $X_{\text{input}}$ is fed into the LLM for auto-regressive decoding. Assuming that the LLM mainly contains two important components: a language decoder $\Gamma$ and an LM head, the loss of conventional multi-modal learning can be expressed as

$$
Loss_{\text{LM}}^{\text{Conv}} = \frac{1}{|S_{\text{LM}}|} \sum_{n \in S_{\text{LM}}} \left( -\sum_{i=0}^{C-1} P_n(i) \log Q_n(i) \right),
\tag{2}
$$
$$
Q = \text{Softmax}(W_{\text{LM}} \Gamma(X_{\text{input}})),
$$

where $W_{\text{LM}}$ is the optimizable parameters of LM head, set $S_{\text{LM}}$ is the set of expected text output index of model, $P$ represents the ground truth label and $C$ is the vocabulary size of model.

However, classical multi-modal learning neglects the supervision function of visual information on multi-modal auto-regressive modelling, which is still multi-modal based language modelling in essence. Our work proposes to leverage the great reasoning potential of LLM to facilitate both language modelling and visual modelling of LMM. To achieve this goal, the key component is to

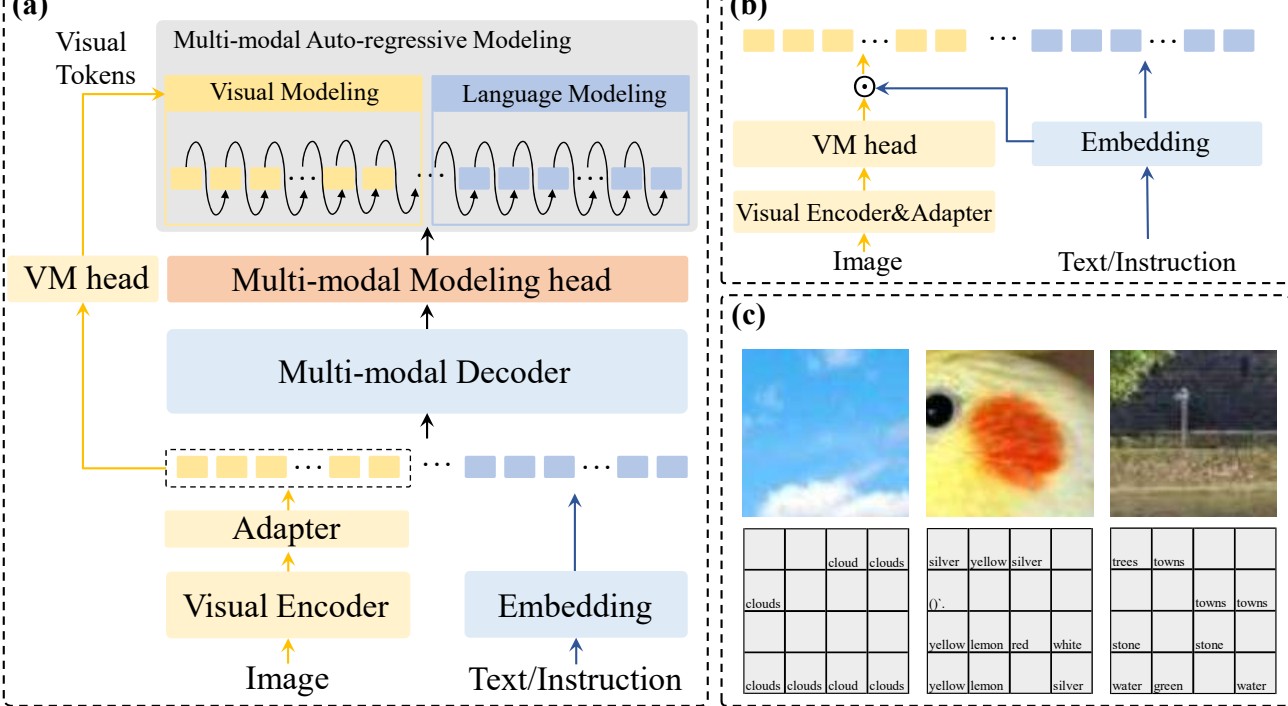

Figure 3: The overview of our method. (a) The overall framework of the model. VT-LMM uses the VM head to transform visual features in multi-modal input sequences to probability distributions over LLM's vocabulary (so-called visual tokens) to participate in visual modelling (b) Constructing pseudo image features with pre-trained embedding of LLM and visual tokens. (c) Demonstration of semantically closest tokens of each image patch in LMM.

construct formally unified multi-modal generative objectives so that the model can be trained with auto-regressive manner. As shown in Figure 3(a), the architecture of VT-LMM consists of five components: a Visual encoder, a multi-modal decoder, an adapter for projection between visual modal and language modal, a Multi-modal Modeling head (MM head) for multi-modal modelling and a corresponding VM head for visual modelling. The embedding module can be regarded as part of multi-modal decoder. We initialise the multi-modal decoder and MM head in VT-LMM using the pre-trained LLM and its LM head. In contrast to classical multi-modal learning, VT-LMM uses VMhead to construct visual tokens for visual features as supervisory information, thus enabling the model to perform multi-modal auto-regressive modeling over the entire sequence.

## 2.2 Visual Tokens

Previous work integrate images and text into a unified structure, enabling the powerful reasoning capabilities of the LLM to generalize from the text space to the multi-modal space. They essentially blur the modality differences in the encoding process of different modalities, as they all participate in the encoding of information in a consistent embedding form.

To further explore the connection between visual features and text embeddings, we search for the token id semantically closest to each image patch. For each feature $x_i \in X_{\text{image}}$, its semantically closest token id $t_i$ can be obtained as:

$$v_i = \arg\min_j \text{Cosine}(x_i, e_j), \quad v_i \in [0, C-1], \quad (3)$$

in which $e_j \in [e_0, e_1, \ldots, e_{C-1}]$ is the pre-trained text embedding of LLM and $\text{Cosine}(a, b)$ is the cosine similarity of vector $a$ and $b$.

We show the semantically closest tokens of image regions in figure 3(c), it can be observed that for the LMM with multi-modal

alignment training, there is a certain correlation between the information contained in each vector of the image features and the actual content of the image. Moreover, this correlation can be explicitly expressed by the vocabulary of the LMM. This finding, on the one hand, proves that the visual features within the LMM are in a similar semantic space as the text embedding, and on the other hand, it provides a new perspective for the interpretability analysis of the LMM. Furthermore, existing LLM methods have shown that the hidden states of text features can be mapped to the model's vocabulary through linear projection, enabling the extraction of interpretable semantics. Therefore, we propose using the linear projection to also map the visual features to the probability distribution over the model's vocabulary, which we called visual tokens, to further strengthen the correlation between visual features and text embeddings.

In this case, the classic language modeling loss can be represented as

$$Loss_{LM} = \frac{1}{|S_{LM}|} \sum_{n \in S_{LM}} \left( -\sum_{i=0}^{C-1} P_n(i) \log Q_n(i) \right), \quad (4)$$

$$Q = \text{Softmax}(W_{MM}\Phi(X_{input})),$$

where $W_{MM}$ is the optimizable parameters of MM head, $\Phi$ represents the multi-modal decoder.

The visual tokens of visual features can be represented as

$$P' = \text{Softmax}(W_{VM}X_{image}), \quad (5)$$

in which $X_{image}$ is the visual embeddings within $X_{input}$ and $W_{VM}$ is the optimizable parameters of VM head. To perform visual modeling on image information using a unified classification format, we design the $Loss_{VM}$ as follows:

$$Loss_{VM} = \frac{1}{|S_{VM}|} \sum_{n \in S_{VM}} \left( \sum_{i=0}^{C-1} P'_n(i) \log(\frac{P'_n(i)}{Q_n(i)}) \right). \quad (6)$$

where set $S_{VM}$ is the set of visual output index of model. It should be noted that in the training process of VT-LMM, the optimisation phase of VM head and multi-modal decoder are separated, and thus do not cause instability during training.

The final optimization objective of multi-modal auto-regressive modeling is represented as

$$Loss_{MM} = Loss_{LM} + Loss_{VM}. \quad (7)$$

Through the additional visual modeling task, we explicitly force the model to capture the distribution of image information in the current semantic space, further eliminating the differences between visual features and text embeddings, thereby improving the model's performance in vision-language comprehension.

## 2.3 Fused Text Embeddings as Pseudo Image Features

The text embeddings in the multi-modal decoder can be regarded as a set of base vectors within its semantic space. Therefore, we propose utilizing the visual tokens and text embeddings to construct pseudo image features, aiming to further explore the manifestation of visual features in the semantic space of the LMM.

As shown in Figure 3(b), we construct the following pseudo image features $X_p$:

$$X_p = W_{embeddings} \odot \text{Softmax}\left( (W_{VM}X_{image}) \right) \quad (8)$$

in which $W_{embeddings}$ represents the weights matrix of multi-modal decoder's pre-trained embeddings, $\odot$ represents the dot product operation.

We replace $X_{image}$ in $X_{input}$ with $X_p$ and Keep loss calculation and other settings unchanged, thereby exploring whether visual information can be seamlessly reconstructed within the language semantic space.

## 3 EXPERIMENTS

### 3.1 Model Pre-training

*3.1.1 Settings.* In the specific implementation, we use CLIP-ViT-L-336px [24] to initialise the visual encoder. for the multi-modal decoder and MM head, we use two different LLM initialisation schemes: the Vicuna-7B [29] and the Mistral-7B [12]. VM head is a randomly initialised unbiased linear layer and adapter is a randomly initialised two-layer MLP. For detailed dataset information and training parameter settings, please refer to table **??**.

*3.1.2 Stage I.* At this stage, we aim to perform the preliminary alignment of the visual information into the semantic space of LLM. To achieve this, we employ adapter to project the uni-modal features output from the visual encoder into the semantic space of the multi-modal decoder, which serves as a contextual reference for the generation task. During this period, training task of VT-LMM is generating image caption for given image. We only train adapter with $Loss_{LM}$ at this stage.

*3.1.3 Stage II.* At this stage, our goal is to adapt the LMM using multi-modal instruction data to obtain an LMM with multi-modal comprehension. At this stage, training task of VT-LMM involves a mixture of tasks, including complex inference, detailed description, multi-round dialogue, image caption, and visual question answering. We train multi-modal decoder, MM head and adapter with $Loss_{LM}$ in this phase.

*3.1.4 Stage III.* In this stage, our objective is to train the VM head using existing LMM and massive image data, enabling it to map visual information to the language semantic space. The training task in this stage is to fit the output of LMM receiving pure image information using the VM head. It is important to note that in this stage, $Q$ serves as the label and $P'$ serves as the logits. i.e., the training loss used in this stage is:

$$Loss'_{VM} = \frac{1}{|S_{VM}|} \sum_{n \in S_{VM}} \left( \sum_{i=0}^{C-1} Q_n(i) \log(\frac{Q_n(i)}{P'_n(i)}) \right). \quad (9)$$

We only train VM head with $Loss'_{VM}$ in this phase.

*3.1.5 Stage IV.* In this stage, our goal is to use the VM head to construct the visual supervision information so as to train the LMM with multi-modal auto-regressive modelling. In this stage, we use the same dataset and training tasks as in Stage 2, but both language and visual information are involved in supervision. We only train the multi-modal decoder and MM head with $Loss_{MM}$ at this stage.

| Methods | LLM | Res. | Visual Question Answering | | | | | Benchmark Toolkit | | | |
|---|---|---|---|---|---|---|---|---|---|---|---|
| | | | VQA$^{v2}$ | GQA | VisWiz | SQA$^I$ | VQA$^T$ | POPE | MMB | MMB$^{CN}$ | MM-Vet |
| *Language Modeling Method* | | | | | | | | | | | |
| IDEFICS-80B [11] | LLaMA-65B | 224 | 60.0 | 45.2 | 36.0 | – | 30.9 | – | 54.5 | 38.1 | – |
| InstructBLIP [4] | Vicuna-13B | 224 | – | 49.5 | 33.4 | 63.1 | 50.7 | 78.9 | – | – | 25.6 |
| BLIP-2 [14] | Vicuna-13B | 224 | 41.0 | 41.0 | 19.6 | 61.0 | 42.5 | 85.3 | – | – | 22.4 |
| LLaVA-v1.5 [19] | Vicuna-13B | 336 | 80.0* | 63.3* | 53.6 | 71.6 | 61.3 | 85.9 | 67.7 | 63.6 | 35.4 |
| InstructBLIP [4] | Vicuna-7B | 224 | – | 49.2 | 34.5 | 60.5 | 50.1 | – | 36 | 23.7 | 26.2 |
| IDEFICS-9B [11] | LLaMA-7B | 224 | 50.9 | 38.4 | 35.5 | – | 25.9 | – | 48.2 | 25.2 | – |
| Qwen-VL [2] | Qwen-7B | 448 | 78.8* | 59.3* | 35.2 | 67.1 | **63.8** | – | 38.2 | 7.4 | – |
| Qwen-VL-Chat [2] | Qwen-7B | 448 | 78.2* | 57.5* | 38.9 | 68.2 | 61.5 | – | 60.6 | 56.7 | – |
| LLaVA-v1.5 [19] | Vicuna-7B | 336 | 78.5* | 62.0* | 50.0 | 66.8 | 58.2 | 85.9 | 64.3 | 58.3 | 30.5 |
| MoE-LLaVA-2.7B×4-Top2 [18] | Phi-2-2.7B | 336 | 77.6* | 61.4* | 43.9 | 68.5 | 51.4 | 86.3 | 65.2 | – | 34.3 |
| *Multi-modal Modeling Method* | | | | | | | | | | | |
| Emu2-Chat [26] | LLaMA-33B | 448 | 84.9* | 65.1* | 54.9 | 65.5* | 66.6* | – | – | – | 48.5 |
| Emu-I [27] | LLaMA-13B | 224 | 62.0 | 46.0 | 38.3 | – | – | – | – | – | 36.3 |
| MM-Interleaved-SFT [28] | Vicuna-13B | 224 | 80.2* | 60.5* | 54.9 | – | 61.0 | – | – | – | – |
| Unified-IO 2 [22] | UIO-2-6.8B | 384 | 79.4* | – | – | **86.2**\* | – | **87.7** | 71.5 | – | – |
| DreamLLM [5] | Vicuna-7B | 224 | 56.6 | – | 38.1 | – | 34.9 | – | – | – | – |
| VL-GPT-I [32] | LLaMA-7B | 224 | 67.2 | 51.5 | 38.9 | – | – | – | – | – | – |
| LaVIT-v2 [13] | LLaMA2-7B | 224 | 68.3 | 47.9 | 41.0 | – | – | – | – | – | – |
| VT-LMM | Vicuna-7B | 336 | 78.9* | 62.7* | 48.3 | 68.1 | 57.6 | 85.9 | 65.9 | 59.8 | 31.3 |
| VT-LMM | Mistral-7B | 336 | **80.8**\* | **65.4**\* | **58.5** | 75.9 | 63.1 | 87.0 | **80.6** | **79.0** | **44.0** |

**Table 1: Comparison among different LMMs on 5 visual question answering benchmarks and 4 benchmark toolkits. Benchmark names are abbreviated due to space limits. VQA-v2 [7]; GQA [10]; VisWiz [8]; SQA$^I$: ScienceQA-IMG [23]; VQA$^T$: TextVQA [25]; POPE [17]; MMB: MMBench [21]; MMB$^{CN}$: MMBench-Chinese [21]; MM-Vet [31]. \*The training images of the datasets are observed during training. The best results and second best results are indicated by boldface and underline, respectively.**

| Hyperparameter | Stage I | Stage II | Stage III | Stage IV |
|---|---|---|---|---|
| batch size | 256 | 128 | 256 | 128 |
| lr | 1e-3 | 2e-5 | 1e-3 | 2e-5 |
| lr schedule | | cosine decay | | |
| lr warmup ratio | | 0.03 | | |
| weight decay | | 0 | | |
| epoch | | 1 | | |
| optimizer | | AdamW | | |
| DeepSpeed stage | 2 | 3 | 2 | 3 |

**Table 2: Training hyper-parameters for the four stages of training.**

| Usage | Source |
|---|---|
| Stage I | LLaVA 1.5-558k |
| Stage II | LLaVA 1.5-mix-665k |
| Stage III | Images of LCS-558K |
| Stage IV | LLaVA 1.5-mix-665k |

**Table 3: Dataset used in training of VT-LMM**

## 3.2 Main Results

To evaluate the effectiveness of our proposed method, we conducted experiments on five widely used multi-modal benchmarks and four LMM benchmark toolkits. The experimental results are shown in Table 1. The experimental results demonstrate that our proposed VT-LMM, by constructing visual modeling tasks consistent with the pre-training tasks format of LLM, guides model to learn both language and visual modalities of the multi-modal sequences in an auto-regressive manner. This further bridges the gap between the two modal features and significantly enhances the model's vision-language understanding capability.

Compared to models of the same scale, VT-LMM demonstrates superior or competitive performance on all evaluation metrics for vision-language understanding. Compared to larger-scale models, VT-LMM, with only the scale of 7B, outperforms some models with scale of 13B and achieves performance similar to models with a parameter scale of 33B. This once again verifies that our method, by introducing visual information supervision, further boosts the multi-modal understanding potential of LMM.

| Methods | LLM | Res. | Adversarial | | | Popular | | | Random | | |
|---------|-----|------|-----|----------|-----|-----|----------|-----|-----|----------|-----|
| | | | Acc | F1-Score | Yes | Acc | F1-Score | Yes | Acc | F1-Score | Yes |
| LLaVA-v1.5[19] | Vicuna-13B | 336 | 85.5 | 84.5 | 43.3 | 87.4 | 86.2 | 41.4 | 88.0 | 87.1 | 41.7 |
| LLaVA-v1.5[19] | Vicuna-7B | 336 | 85.1 | 84.2 | 44.0 | 87.2 | 86.1 | 41.9 | 88.1 | 87.3 | 41.9 |
| mPLUG-Owl [30] | LLaMA-7B | 224 | 82.4 | 81.6 | 45.2 | 85.5 | 84.3 | 42.1 | 86.3 | 85.3 | 42.3 |
| Multimodal-GPT [6] | LLaMA-7B | 224 | 50.0 | 66.7 | 100.0 | 50.0 | 66.7 | 100.0 | 50.0 | 66.7 | 100.0 |
| MoE-LLaVA-2.7B×4-Top2 [18] | Phi-2-2.7B | 336 | 85.9 | 84.9 | 43.2 | 87.5 | 86.4 | 41.8 | 88.5 | 87.7 | 41.8 |
| VT-LMM | Vicuna-7B | 336 | 85.1 | 84.1 | 44.0 | 87.4 | 86.3 | 41.7 | 88.1 | 87.3 | 41.9 |
| VT-LMM | Mistral-7B | 336 | **87.3** | **86.1** | 41.9 | **88.3** | **87.1** | 40.9 | **88.7** | **87.8** | 41.4 |

**Table 4: Results of object hallucination evaluation. Yes denotes the proportion of answering "Yes" to the given question. The best results and second best results are indicated by boldface and underline, respectively.**

| Methods | LLM | Res. | Visual Question Answering | | | | | Benchmark Toolkit | | | |
|---------|-----|------|-----------|-----|--------|------|------|-------|-----|-------|--------|
| | | | $VQA^{v2}$ | GQA | VisWiz | $SQA^I$ | $VQA^T$ | POPE | MMB | $MMB^{CN}$ | MM-Vet |
| VT-LMM | Vicuna-7B | 336 | 78.9 | 62.7 | 48.3 | 68.1 | 57.6 | 85.9 | 65.9 | 59.8 | 31.3 |
|   - visual modeling | Vicuna-7B | 336 | 78.5 | 62.0 | 50.0 | 66.8 | 58.2 | 85.9 | 64.3 | 58.3 | 30.5 |
| VT-LMM | Mistral-7B | 336 | 80.8 | 65.4 | 58.5 | 75.9 | 63.1 | 87.0 | 80.6 | 79.0 | 44.0 |
|   - visual modeling | Mistral-7B | 336 | 79.1 | 62.5 | 52.6 | 72.4 | 56.6 | 87.1 | 70.0 | 63.6 | 36.3 |
| VT-LMM with pseudo image features | Vicuna-7B | 336 | 77.2 | 61.6 | 48.1 | 65.4 | 54.5 | 85.6 | 65.2 | 53.5 | 30.1 |

**Table 5: Results of ablation study and discussion.**

Compared to methods that solely apply language modeling as the optimization task, VT-LMM achieves significantly superior results, highlighting the importance of introducing visual insformation as supervision for LMMs. Existing methods that also employ multimodal modeling tasks and perceive the supervision of visual information can be broadly classified into two categories. One involves constructing an additional image decoder to endow the model with image generation capabilities, thereby utilizing image information in the form of image denoising tasks. The other models the image information through regression tasks, aiming to directly fit the value of visual features using MSE loss. In comparison to the above methods, VT-LMM achieves better results by constructing visual tokens as supervision information, enabling a relatively unified and concise implementation of multi-modal modeling in the form of a classification task. This approach proves to be more effective and efficient.

It should be noted that both LaVIT and VT-LMM employ classification task for multi-modal modeling. However, LaVIT introduces an additional visual tokenizer and 16,384 trainable discrete embeddings. In contrast, visual tokens utilize the existing semantic space of the pre-trained LLM to represent visual information, avoiding the introduction of massive training parameters. This approach also bridges the semantic gap between the two modalities and thus achieves better results.

### 3.3 Object Hallucination Evaluation

We adopt the Polling- based Object Probing Evaluation (POPE) to evaluate object hallucination in VT-LMM and report results

in table 4. The two variants of VT-LMM have achieved excellent performance in models of the same scale. Notably, VT-LMM-Mistral-7B has achieved the best performance across three different data sampling methods: Adversarial, Popular, and Random. Additionally, the yes ratio of VT-LMM remains relatively balanced, indicating that our method can provide accurate identification results based on the given query. These results demonstrate that our VT-LMM, achieving unified multi-modal auto-regressive modeling across multi-modal sequences by introducing visual tokens, explicitly forces the model to learn the semantic distributions of both visual and language information, leading to improved visual-language consistency. i.e., The model tends to perceive and generate object information that aligns with the given image content.

### 3.4 Ablation

In order to investigate whether the introduction of visual tokens as visual supervision information directly improves the vision-language comprehension performance of LMMs, we conducted an ablation study, and the experimental results refer to 5. From table 5, it can be observed that the introduction of visual tokens leads to evident performance enhancement in both different LLM settings, which confirms that VT-LMM, by constructing visual tokens as visual supervision information, guides the model to learn the rich semantics in multi-modal sequences through both language modelling and visual modelling, therefore boosting model's vision-language understanding capability. In addition, we note that VT-LMM-Vicuna-7B shows a slight performance degradation on both VisWiz and TextVQA benchmarks. This case is due to the

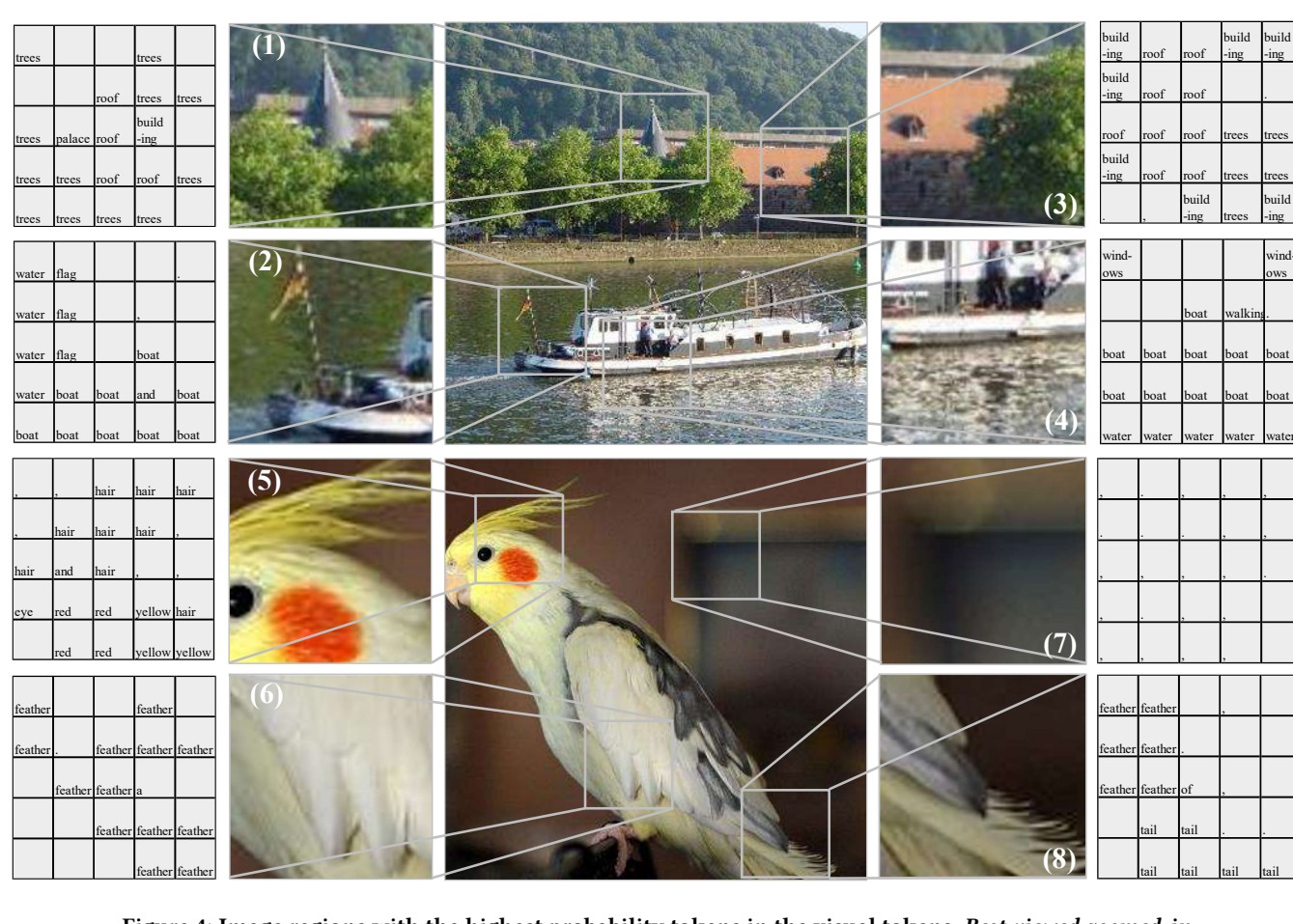

**Figure 4: Image regions with the highest probability tokens in the visual tokens.** *Best viewed zoomed-in.*

fact that VisWiz involves the requirement of the corresponding format of the model and TextVQA provides a large number of image OCR texts as reference. In both tasks, the language semantic understanding and instruction following capabilities of the model are more demanded. And the introduction of the visual modelling task slightly diminishes the text-side capabilities of Vicuna-7B on both datasets. When using Mistral-7B as the LLM, its stronger baseline performance and pre-trained language semantic comprehension bridged this gap, which in turn led to a consistent performance improvement.

## 4 DISCUSSION

### 4.1 Replace Visual Feature with Pseudo Image Features

To further explore the manifestation of visual features in the semantic space of the LMM, we utilize the visual tokens and text embeddings to construct pseudo image features and replace visual features in the input sequence. The results are shown in table 5. From the experimental results, it is clear that the model that receives pseudo image features as visual input can still achieve vision-language understanding capability close to the original model. This result

confirms that the embedding layer of the pre-trained LLM essentially achieves full utilisation of the semantic space, thus features from image modalities can be represented by linear combinations of text embeddings to some extent. Moreover, it also shows that visual tokens successfully implement the transfer of visual features to the language semantic space, again verifying the soundness of our proposed approach. However, the difference in results suggests that the structure of the linear projection might have limitations for the projection of visual information into the language semantic space, and we will further investigate how to better accomplish this operation in our subsequent work.

### 4.2 Visualization of Visual Tokens

In order to verify whether the visual tokens learnt by VT-LMM can realistically reflect the image information, we take VT-LMM-Vicuna-7B as an example to explore. For each patch in the image, we select the token with the highest probability in its corresponding visual tokens, and compare the region of interest in the image with its visualisation result, visualization is shown in figure 4.

From the visualisation results, it can be observed that: First of all, visual tokens can intuitively perceive the low-level features of images such as colour information and object boundary contours.

The model in region (5) identifies the red region of the parrot's face; the distribution of visual tokens in regions (5), (6) (8) clearly distinguishes the boundaries of the foreground and the background; and the visual tokens in regions (1), (3) accurately identifies the demarcation line between the building and the tree.

Secondly, visual tokens are able to recognise higher level semantics such as specific object categories. For example, roof, building, and tree are successfully recognised in regions (1) and (3); flag and window are recognised in regions (2) and (4); hair and feather of parrot are recognised in regions (5), (6), and (8); in region (4), visual tokens even mark the "walking" action of the crew.

In addition, we found that for the relationship between foreground and background in the internal semantic space of LMM, the model prefers to predict the background as a comma rather than a space or other placeholder as we expected. This phenomenon reflects the fact that visual tokens are simultaneously characterised by both visual and textual information, i.e. the background is seen in the image as a separator between different foreground patches.

The above phenomena indicate that visual tokens successfully achieve the transformation of visual features to language semantic space, which verifies the feasibility and effectiveness of our method. Therefore, using visual tokens as visual supervision signals can indeed guide the model to perform auto-regressive modelling of visual information, thus enhancing the model's multi-modal comprehension and inference capabilities.

## 5 RELATED WORK

### 5.1 Vision-Language Pre-training

Vision-Language Pre-training aims to build diverse multi-modal contextual training methods, thus endowing models with stronger multi-modal comprehension. Existing work has performed extensive and thorough research on Vision-Language Pre-training (VLP). CLIP [24] proposes to apply contrastive learning with both visual encoder and text encoder to learn generic cross-modal representations, which lays the foundation of the learning paradigm for VLP. ALBEF [16] further improves the learning effectiveness of VLP by applying Image-Text Contrastive loss (ITC) and Image-Text Matching task (ITM) to the classical Masked Language Modelling (MLM) task to further enhance the learning of cross-modal representations. BLIP [15] combines ITM, ITC and language modelling task (LM) as a classical work on multi-modal generative training methods. With the rapid development of LLM, researchers begin to explore LLM applied to vision-language tasks. Flamingo [1] uses interleaved text-image data for training models with open generative capabilities; BLIP-2 [14] and InstructBLIP [4] use contrastive learning and ML tasks to construct efficient visual resampler for LLM at a low price. KOSMOS [9] proposed to align the visual perception and language from scratch on web-scale multi-modal corpora. Previous VLPs used image-text pairs datasets and simple VQA datasets with limited data diversity and task complexity to enhance the general understanding and generation of Large Multi-modal Model (LMM). With the progress of instruction-following in LLM, Multi-modal instruction-following task is proposed. LLaVA series [19, 20] construct complex Multi-modal instruction-following data and significantly improves model's visual comprehension, and can be effectively extended to other vision-language tasks such as text understanding and region dialogue.

However, the visual inputs in the above approaches are only considered as hints for generating targets and are not involved in the optimisation, which severely limits the model's potential for multi-modal comprehension and results in incomplete utilisation of VLP training data.

### 5.2 Visual Supervised Information

To further leverage visual information for supervised learning, Emu [27] and Emu2 [26] align visual and language modelling with the objective of predicting the next visual or language token in an auto-regressive manner, and further explore video as a new source of interleaved image-text data. This unified modelling provides a general framework for multi-modal understanding and generation. LaVIT [13] utilizes narest neighbor lookup to map image information to discrete trainable codebooks, and extends the original vocabulary of LLM to construct one-hot supervised labels for image information. Unified-IO 2 [22] improves the modeling capability of the backbone for multi-modal information by predicting corrupt modalities using an additional VQ-GAN decoder and optimizing the pixel-level reconstruction loss.

However, the above methods either treat visual modelling as regression task, which does not match the pre-training format of LLM, or need to introduce extensive external training parameters and complex structures. In contrast to these methods, our approach uses LLM's own word embedding space to construct visual tokens as supervised signals, using the classification task for visual modelling and do not need to introduce additional visual embedding.

## 6 CONCLUSION

In this paper, we successfully perform multi-modal auto-regressive modeling with a unified objective for the first time. Specifically, we propose the concept of visual tokens which transforms visual features in multi-modal sequences into probability distributions over the vocabulary of pre-trained LLMs, thus constructing supervision label for visual modeling. We also verified the representation of visual information by visual tokens and the feasibility of using visual tokens together with pre-trained embeddings to represent visual information in the language semantic space. The results show that visual tokens successfully achieve vision2language semantic transformation and effectively enhance the vision-language comprehension of model.

## 7 LIMITATIONS

Mapping continuous visual information into discrete language semantic space leads to a certain degree of information loss. This is also evidenced by the slight performance degradation brought about by the use of pseudo image feature. Furthermore, the effect of the diversity of training images on the learning effectiveness of the VM head was not explored. Therefore, it is necessary to further explore a more appropriate structure for constructing visual tokens and to verify the learning effect of VM head using richer image data.

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
