# OpenReview forum: "Multi-modal Auto-regressive Modeling via Visual Tokens"
_acmmm.org/ACMMM/2024/Conference — MM2024 Poster_

### Official Review · Reviewer_vBdQ · 2024-05-25

**Rating:** 3
**Confidence:** 3

**Summary:**

This paper proposes an approach to align the visual features with text embeddings in LLM. It introduces the term "visual tokens" from the semantic space in LLM. The method gets the visual label from text embeddings and does the autoregressive in multi-modal decoder.

**Strengths:**

The authors' proposition to utilize labels from a large language model dictionary as representatives for the labels of visual patches showcases a degree of innovativeness.
They extensively validate their approach through a multitude of experiments, achieving competitively advantageous results across various tasks.
The motivation behind the study is articulated clearly, and the authors' writing is commendable.

**Limitations:**

1.I don't think this method of obtaining labels as reasonable, even though it yields decent results. I argue that comparing the similarity between visual features and text embeddings isn't appropriate. In CLIP, we derive embeddings from a text dictionary, feed them into a transformer-based text encoder to get text features, and then contrast these with visual features for learning. Thus, I believe we shouldn't forcibly align visual features with text embeddings. An experiment could be conducted where visual features of the same class, despite being different, are examined to see if they can map to the same text embedding.

2.The authors haven't clarified whether the LLM (Large Language Model) is frozen during training. If it is frozen, the obtained labels may not necessarily represent the semantically closest text embeddings. Since LLM's text embeddings lack visual context awareness while LMM's might be more suitable, I expect a sound explanation from the authors regarding this aspect.

3. There are some writing inconsistencies; the usage of LLM and LMM appears confusing and might contain errors. It would be beneficial to thoroughly proofread and correct these instances.

4.Table L428 seems to be missing, which could be a typographical error or an oversight that needs addressing.

**Suitability:**

3

---

### Official Review · Reviewer_7DC4 · 2024-06-01

**Rating:** 3
**Confidence:** 2

**Summary:**

This paper proposes a multi-modal auto-regressive method for training large multi-modal models. Compared to existing methods that typically map visual features into the LLM space and then predict text tokens only, the proposed pipeline uses visual tokens to construct visual supervision labels, enabling multi-modal auto-regressive modeling with a unified classification objective.

**Strengths:**

- Extensive experiments were conducted and achieved performance comparable to state-of-the-art methods.
- An interesting trial incorporates visual features to enable multi-modal auto-regressive modeling with a unified classification objective.
- The visualization of learnt visual tokens seems to demonstrate that the method can effectively learn to map image patches to the most semantically closest concept.

**Limitations:**

- 'Visual Token' is a widely used concept in multi-modal large language models, so it is confusing when the authors claim they are proposing the concept of visual tokens. Additionally, it is perplexing that the proposed framework focuses solely on the image-to-text task, such as VQA. Why is it necessary to predict so-called visual tokens if they are not used to generate image content? The concept of visual tokens here is misleading.
- The introduction section is not clear enough, especially regarding the differences between 'mainstream methods,' 'visual regression methods,' and 'multi-modal auto-regressive methods.' Additionally, 'mainstream methods' is not a specific type of method; it would be better to provide a specific type name.
- Too complicated training pipeline: while existing alignment methods like LLaVA and MiniGPT require only one or two stages to map the image features to the language space by training on relatively small alignment datasets, the proposed method requires four training stages, which is not efficient.
- Some references are missing, the authors should carefully proofread their manuscript, e.g., line 428, table??; line 686, refer to 5 -> refer to Table 5;
- Some mathematical expressions and their explanations are confusing. For example, in Section 2.2, Equation (3) mentions that the closest token ID $t_i$ can be obtained as, .... However, $t_i$ is not shown in Equation (3); instead, a definition for $v_i$ is given. Also, it is not clear what is meant by "we search for the token ID semantically closest to each image patch." What is the 'token' here? Is it a text token, a visual token, or something else? The description of the method is not clear enough.
- Since this work still uses a similar linear projection technique as existing methods (referred to as the 'mainstream method' in this paper) to map visual features into the LLM space, it primarily introduces additional steps or features to enhance existing frameworks. However, how can we balance the trade-offs? This approach also introduces extra training stages and computational complexity.

**Suitability:**

3

---

### Official Review · Reviewer_QYHQ · 2024-06-02

**Rating:** 4
**Confidence:** 3

**Summary:**

The paper "Multi-modal Auto-regressive Modeling via Visual Tokens" explores the extension of Large Language Models (LLMs) to multi-modal scenarios, leading to the development of Large Multi-modal Models (LMMs). The authors introduce the concept of visual tokens, which map visual features to the language model's vocabulary, allowing for unified multi-modal auto-regressive modeling. This approach leverages both visual and textual information for supervised learning, enhancing the model's performance on visual question answering tasks. Experimental results demonstrate the model's superior or competitive performance compared to other state-of-the-art models.

**Strengths:**

1. **Innovative Use of Visual Tokens**: The introduction of visual tokens, which map visual features to the language model's vocabulary, is a innovative approach. This allows for the unified modeling of both visual and textual information, leveraging the strengths of pre-trained large language models (LLMs) without requiring extensive architectural changes .
2. **Ablation Studies and Visualizations**: The paper includes detailed ablation studies and visualizations of the visual tokens, providing clear evidence of how visual features are integrated and utilized within the language semantic space.

**Limitations:**

1. Inline reference missing in line 428.

2. Impact on Text-Side Capabilities: The introduction of the visual modeling task has been observed to slightly diminish the text-side capabilities of the base language models (e.g., Vicuna-7B). This indicates a trade-off between visual and textual performance that needs to be balanced .

3. Accuracy of Mapping Between Image Tokens and Text Tokens: In Figure 4 and Equation (3), the authors observed a certain correlation between image tokens and input text tokens. However, I would appreciate seeing quantitative results that demonstrate the accuracy of this mapping. Such data would provide a more concrete understanding of how effectively the model translates visual information into textual representations.

4. Missing of Important Baselines: It is recommended that the authors include comparisons with more recent state-of-the-art (SOTA) methods, such as LLaVA-Next. Specifically, LLaVA-Next, utilizing the same base language models (LLMs) like Vicuna-7B and Mistral-7B, demonstrates superior performance across multiple benchmark datasets compared to the authors' proposed method.

| Method      | LLM         | Act.  | SQA IMG | Text VQA | GQA  | POPE | MME  | MMB EN | MMB CN | MM Vet | VQA v2 | LLaVA Wild | SEED IMG | MMMU val | Math Vista |
|-------------|--------------|-------|---------|----------|------|------|------|--------|--------|--------|---------|------------|----------|-----------|-------------|
| LLaVA-NeXT  | Mistral-7B   | 7.6B  | 72.8    | 65.7     | 64.8 | 86.7 | 1498 | 68.7   | 61.2   | 47.3   | 82.2    | 83.2 | 72.2       | 35.3     | 37.7       |
| LLaVA-NeXT  | Vicuna-7B    | 7.1B  | 70.1    | 64.9     | 64.2 | 86.5 | 1519 | 67.4   | 60.6   | 43.9   | 81.8    | 81.6 |  70.2       | 35.8     | 34.6       |

[Table obtained from Li, Jiachen, et al. "CuMo: Scaling Multimodal LLM with Co-Upcycled Mixture-of-Experts." arXiv preprint arXiv:2405.05949 (2024).]

**Suitability:**

3

---

### Official Review · Reviewer_KQne · 2024-06-04

**Rating:** 4
**Confidence:** 3

**Summary:**

The paper presents a novel approach to multi-modal auto-regressive modeling by introducing the concept of visual tokens. These tokens map visual features into the language semantic space, enabling Large Multi-modal Models (LMMs) to perform auto-regressive modeling over multi-modal sequences. The proposed method demonstrates superior performance on several visual question answering (VQA) benchmarks and achieves competitive results compared to larger models.

**Strengths:**

(1)The introduction of visual tokens to map visual features into the language semantic space is innovative.

(2)The approach unifies visual and language modeling, enhancing the potential of LMMs.()

(3)The method shows promising results, outperforming several existing models of similar or larger scale.

**Limitations:**

(1)The use of pseudo image features leads to a noticeable performance drop, which suggests limitations in the current implementation.

(2)The training process is quite complex, divided into four stages, with the dataset being used twice. The performance improvement may be attributed to the repeated use of the data. The overall logic of the paper seems a bit disorganized. Using a single figure to represent the four stages of training might make it unclear.

(3)There is an extra punctuation mark on line 167.

(4)There is a problem with the hyperlink in the table at line 428.

(5)For section 3.1, I believe it is more appropriate to refer to the pre-train stage as 'instructional fine-tuning'.

(6)Throwing a visual token into the text embedding to get the same word as the text is somewhat counterintuitive.

**Suitability:**

3

---

### Official Review · Reviewer_XvkL · 2024-06-06

**Rating:** 4
**Confidence:** 2

**Summary:**

The paper tackles the challenge of creating LMM by extending auto-regressive modeling to include images. The authors propose "visual tokens" to map visual features to the model's vocabulary, providing the needed supervision for visual data. They also explore using text embeddings to represent visual information. Lastly, experimental validation results show that this method performs effectively.

**Strengths:**

1) The author claims that it is the first work that proposes multi-modal auto-regressive modeling with unified objectives.
2)  The paper is also commendable for its comprehensive evaluation, validated through extensive experiments on five VQA tasks and four benchmark toolkits, underscoring the proposed method's robustness and generalizability.
3) Also, the paper presenting complex concepts like visual tokens in a clear and logical manner, making the methodology easy to understand and replicate.

**Limitations:**

1) Equation 1 isn't it just repeating the same thing as shown in the orange boxes in Figure 2or3? Please point out if I misunderstood; if not, I guess there is a better way to improve the clarity by replacing the texts in Figure 2or3 with the symbol/variable names in Equation 1; this could save some space to insert more details of this proposed method.
2) For Figure 3c, can you explain more about the wordings or symbols in each column? Not so clear for me, why is there a 0 ' ?
3)  In the results table, Table 1,4,5, is the 'res' means resolution here? Can you analyse and compare with Unified-IO 2 since they are also having good results in comparable res.? The VT-LMM results shows the best only when you are not comparing to [22] method so I am quite curious on the comparison between these two works.
4) Small comment on your punctuation. Please keep them the same across the whole paper, for example, table in capital T or not.

**Suitability:**

3

---

### Meta-Review · Area_Chair_pYrJ · 2024-06-29

**Recommendation:** Accept (Poster)
**Confidence:** 4

**Metareview:**

The paper received four positive ratings and one negative rating, placing it on the borderline. Most reviewers appreciated the novelty of the paper, which maps visual features to probability distributions over the LLM’s vocabulary, providing visual supervision for unified multimodal autoregressive modeling. The AC agrees and encourages any work that advances multimodal large models.

However, reviewers pointed out that the writing is poor and needs significant improvement. They also expect more quantitative experimental results to make the paper more convincing, with which the AC also agrees.

Overall, the AC believes that the pros outweigh the cons and thus recommends acceptance.